# Consumer Preferences and Willingness to Pay for High-Quality Vegetable Oils: A Cross-Sectional Analysis of Chinese Residents

**DOI:** 10.3390/foods13081168

**Published:** 2024-04-11

**Authors:** Shiting Tang, Pei Wang, Huan Xing, Zeying Huang, Peng Liu, Tai Li, Jiazhang Huang

**Affiliations:** 1Institute of Food and Nutrition Development, Ministry of Agriculture and Rural Affairs, Beijing 100081, China; 82101215562@caas.cn (S.T.); 82101225640@caas.cn (H.X.); huangzeying@caas.cn (Z.H.); liupeng@caas.cn (P.L.); litai1010@163.com (T.L.); 2Key Laboratory of Environmental Medicine and Engineering of Ministry of Education, Department of Nutrition and Food Hygiene, School of Public Health, Southeast University, Nanjing 210009, China; wangpei929@seu.edu.cn

**Keywords:** vegetable oil, preferences, willingness to pay, consumers, TPB, structural equation model

## Abstract

The consumption of edible vegetable oil is an important source of essential fatty acids and vitamin E for the human body. Guiding residents to consume scientifically and reasonably control the intake of edible oils is an important part of promoting the construction of a healthy China. Currently, Chinese residents have an insufficient understanding of the scientific consumption of edible oils, leading to an intake exceeding the dietary recommendations, resulting in excessive fat intake and increasing the risk of chronic diseases such as obesity and cardiovascular diseases. Based on the Theory of Planned Behavior (TPB) and using Structural Equation Modeling (SEM), this study analyzed the cognitive preferences and willingness to pay a premium for edible vegetable oils among urban consumers in China. The survey included 1098 Chinese urban consumers of vegetable oils. The research found that attitudes, health value, monetary value, and environmental endowment of urban residents are the main predicting factors of the intention to purchase high-quality vegetable oils. This study confirms the applicability of the Theory of Planned Behavior in the consumption of high-quality vegetable oils and provides theoretical contributions and insights for Chinese enterprises and policymakers in formulating marketing strategies for vegetable oils.

## 1. Introduction

The consumption of edible vegetable oil is indispensable for human beings, as it is rich in fats which are one of the three major nutrients needed by the human body, and also provide various other nutrients that play a very important role in human health [1]. At the same time, the healthy consumption of edible vegetable oil is an important indicator of the living standards of a country’s people [2].

China is a major consumer of edible vegetable oil, and in recent years, the consumption of vegetable oil has been growing rapidly [3]. Currently, China’s consumption of vegetable oil ranks first in the world. According to statistics, the per capita consumption of vegetable oil in China was 15.3 g/d in 1990, and it reached 69.8 g/d in 2020, representing a 437.8% increase in calorie intake compared to 1990 [4]. The rapid growth in the consumption of vegetable oil in China over the past 30 years has played a positive role in improving the living standards and dietary nutrition of residents during certain historical periods. However, due to insufficient understanding of the scientific intake of vegetable oil, there are prominent issues in current consumption habits among Chinese residents, such as excessive consumption and a limited variety of types. According to the China Resident Nutrition and Health Status Survey conducted in 2017, the average daily intake of cooking oil per capita in Chinese households was 43.2 g, exceeding the recommended upper limit (30 g) by 13.2 g.

The unreasonable consumption of edible oils not only increases health risks for urban and rural residents, but also places a significant burden on the national oil crop production and oil supply, leading to heavy social burdens [5]. Excessive consumption of vegetable oils and fats can also increase the risk of overweight, obesity, cardiovascular diseases, and other diseases. The “Report on Nutrition and Chronic Disease Status of Chinese Residents (2020)” released by the National Health Commission of China shows that the problem of overweight and obesity among residents is becoming increasingly prominent, and the prevalence and incidence of chronic diseases are still on the rise [6].

Evidence-based studies show that the incidence of obesity, “three highs” (hyperlipidemia, hypertension, and hyperglycemia), and cardiovascular diseases among residents is increasing year by year, and these diseases are closely related to unhealthy dietary habits, including the consumption of cooking oil [7,8]. The World Health Organization (WHO) has released dietary guidelines based on scientific research evidence, which suggest that reducing the intake of saturated fatty acids and trans fatty acids is associated with a lower risk of cardiovascular diseases and overall mortality. It is recommended to use polyunsaturated fatty acids and monounsaturated fatty acids from plant sources. Therefore, human health is intricately linked not just to the consumption of vegetable oils but to the broader context of total fat intake, lifestyle, physical activity, and dietary habits. It is essential for individuals to focus on the informed consumption of vegetable oils, manage their overall fat intake wisely, and adopt healthy lifestyle practices.

The quality of vegetable oils is a matter of great concern for consumers and is also one of the important factors that residents consider when choosing edible oils. Previous studies have evaluated the quality of vegetable oils in terms of safety [9,10,11], nutritional value [12], extraction methods [13,14,15], and health benefits [16,17]. Although there is no clear definition of high-quality vegetable oils in the academic community, it is generally believed that they should meet national food safety standards, have high nutritional value and health benefits, such as reasonable fatty acid composition, be rich in vitamin E, plant sterols, and other trace nutrients and functional active substances, and demonstrate stable product quality and good flavor. Providing healthy foods, including high-quality vegetable oils, is one of the main challenges society faces in terms of diet [18]. Studying consumer preferences for high-quality vegetable oils and their willingness to pay can help guide residents to make rational consumption choices and provide references for enterprise production.

Through our study, we aim to answer two core questions: (1) What attributes and qualities do consumers perceive as necessary for high-quality vegetable oil? (2) What factors influence consumers’ willingness to pay for high-quality vegetable oil? This article is one of the few studies focusing on the willingness of consumers in developing countries to pay for vegetable oils. By considering consumers’ nutritional and health needs, this research can help businesses develop vegetable oil products with higher health value and conduct targeted health promotion. Additionally, this study marks the first attempt to test consumers’ perceived value of vegetable oil in one of the largest vegetable oil-producing countries.

## 2. Theoretical Substantiation and Research Hypotheses

### 2.1. Theoretical Substantiation

In the previous literature on food consumption research, the factors related to consumer preferences and purchase intentions for food were mainly explored using Structural Equation Modeling, including aspects such as behavior attitude, subjective behavior norms, and environmental endowments. The Theory of Planned Behavior (TPB) points out that behavior attitude, subjective behavior norms, and environmental endowments are the main factors influencing willingness to pay. Some studies have incorporated social demographic factors. Tekla Izsó and Barbara Szabó-Bódi used descriptive statistical methods and partial least squares Structural Equation Modeling to demonstrate that the perceived price–value ratio of a product, along with respondents’ cooking skills, knowledge, awareness, and general preference for sour cream, significantly influence the willingness to purchase sour cream substitutes [19]. M.L. Mitterer-Daltoé used Structural Equation Modeling (SEM) to explain the relationship between attitude, subjective norms, and past experience with the fish consumption behavior of Brazilians [20]. Shu-Yen Hsu and others applied Structural Equation Modeling to show that subjective knowledge of organic food, health consciousness, and concerns about food safety have a positive and significant impact on willingness to purchase organic food [21]. Structural Equation Modeling (SEM) technology can simultaneously use different variables and test their interrelationships. This technique examines the structural relationships expressed in a series of multivariate regression equations.

In studies on the consumption and purchase of vegetable oil, existing research primarily focuses on consumers’ cognition, preferences, purchase intentions, and behaviors regarding specific vegetable oils. From different perspectives, the influencing factors that consumers consider when selecting and evaluating specific vegetable oils are analyzed. These factors include health and safety, sustainability, taste, brand, and quality, among others.

In the consumption of vegetable oil, Fabio Verneau used Structural Equation Modeling to demonstrate that health concerns are the main driving factor in reducing willingness to consume palm oil, and consumers’ attitudes towards the environment and social fairness directly and significantly impact their willingness [22]. Anita Silvana Ilak Peršurić employed a mediation model to investigate the role of health behavior in olive oil consumption motivation, as well as the mediating role of perceived health benefits of olive oil in this relationship [23]. Maria Elena Latino and others conducted a systematic literature review, describing and analyzing 47 studies published in the past 20 years, discussing the effects of origin, sustainability, brand, health and safety, production process, packaging, color, taste and flavor, and product characteristics on olive oil [24]. Giuseppe Di Vita used cluster analysis to study the impact of different qualities of olive oil on consumers [25]. The method of Structural Equation Modeling used in existing studies is worth learning from. Previous studies mostly focused on specific vegetable oils without investigating consumers’ overall awareness and willingness to pay for vegetable oil. This study uses Structural Equation Modeling to research consumers’ cognition, preferences, and willingness to pay for vegetable oil, explaining the relationships between consumers’ willingness to consume vegetable oil and filling the research gap in this area. This provides theoretical support for guiding vegetable oil consumption and promoting the high-quality development of oilseed industries.

### 2.2. Research Hypotheses

From a psychological perspective, the process of urban residents’ willingness to pay a premium for high-quality vegetable oil is complex and variable [26,27,28]. The Theory of Planned Behavior has strong explanatory power for the formation process of individual behavioral intentions and can comprehensively reflect the psychological and behavioral aspects of urban residents’ willingness to pay a premium for high-quality vegetable oil. Based on the Theory of Planned Behavior, this study hypothesizes that consumers’ willingness to pay a premium for high-quality vegetable oil is influenced by four factors: perceived value, behavioral attitude, subjective behavioral norms, and perceived behavioral norms. Additionally, these four factors are also affected by internal and external factors such as personal attitudes, resource environment, and social environment. Accordingly, the following research hypotheses are proposed:

#### 2.2.1. Health Value

Each researcher has different perspectives on perceived value. According to Zeithaml, perceived value is defined as the evaluation by consumers of the benefits of a product, especially in comparison to the sacrifices they have to make, and the value they perceive or receive [29]. Zeithaml believes that perceived value is the result of the interaction of monetary and non-monetary factors [29]. Holbrook and Corfman proposed views on values, which are seen as context-dependent, depending on the background in which evaluative judgments occur [30]. This helps to explain the diversity of the meaning of value. Perceived value influences consumers’ attitudes [31]. Oh’s research on perceived value shows a strong relationship between consumers’ willingness and perceived value when choosing a restaurant [32]. Other studies indicate that perceived value also influences intention prediction [33].

Research results suggest that the main factors influencing consumers’ willingness to pay are health-related factors [34,35,36]. Recent studies have incorporated consumers’ perceived health value as a determinant of food purchase intentions and behaviors [37]. The perceived health value by consumers is crucial in the food market, especially for high-quality agricultural products [38].

Based on the existing literature, this study proposes the following hypotheses:

**Hypothesis (H1).** 
*Health value has a positive influence on the willingness to pay a premium for high-quality vegetable oil.*


#### 2.2.2. Monetary Value

When it comes to the willingness to pay a premium for high-quality vegetable oil, consumers typically consider multiple factors. Firstly, the raw material costs for high-quality vegetable oil are usually higher because premium ingredients come at a higher price. Secondly, the processing technology for high-quality vegetable oil is also more complex and refined, which adds to the costs. Additionally, consumers have higher demands for the quality, taste, nutritional content, etc., of the product, making them willing to pay a higher price. Based on Zeithaml’s perspective [29], perceived value is the result of the interaction between monetary and non-monetary factors [39], as pointed out by Shuhaiber [40]. The literature indicates that monetary value is the strongest motivator for purchasing organic food [41], and the perception that food offers value for money is positively associated with willingness to pay [42]. Therefore, residents’ perception of the value of their money may influence their willingness to pay a premium [34,35,36].

**Hypothesis (H2).** 
*Monetary value has a positive influence on the willingness to pay a premium for high-quality vegetable oil.*


#### 2.2.3. Normative Beliefs

Subjective norms refer to the social pressures perceived by urban residents when forming their willingness to pay a premium for high-quality vegetable oil. According to the Theory of Planned Behavior, individual subjective norms are the combination of individual normative beliefs multiplied by individual compliance motivations [43].

In terms of food choice, considering the “significant others” including family members, friends, relatives, colleagues, and other reference groups’ preferences is important [44]. Moreover, Zagata found that normative beliefs enhance consumers’ willingness to purchase organic food, with the most critical source being the family, such as partners or other household members [45]. Therefore, this study proposes the following hypotheses:

**Hypothesis (H3).** 
*Normative beliefs have a positive influence on the willingness to pay a premium for high-quality vegetable oil.*


#### 2.2.4. Compliance Motivation

Consumer compliance motives for high-quality vegetable oil, namely their tendency to adhere to the normative beliefs of significant others regarding high-quality vegetable oil [46], play a significant role in willingness to pay a premium [47], in a thorough analysis of residents’ charitable donation behavior, explicitly pointed out that normative beliefs need to be combined with compliance motives to form actual subjective behavioral norms. Therefore, when consumers consider whether they are willing to pay a premium for vegetable oil, their subjective behavioral norms actually stem from the degree of normative beliefs held by their compliance with friends, other significant individuals, or organizational groups. Subjective behavioral norms are the product of the accumulation of normative beliefs and compliance motives. Scalco [44] also found that compliance motives have a significant impact on the willingness to purchase organic food. This further emphasizes the importance of compliance motives in consumer behavioral decision making. Therefore, this study proposes the following hypotheses:

**Hypothesis (H4).** 
*Compliance motivation has a positive influence on the willingness to pay a premium for high-quality vegetable oil.*


#### 2.2.5. Environmental Endowment

Perceived behavioral control is an individual’s perception of whether they have all the available means and opportunities to perform a behavior [48]. It refers to consumers’ perception of the difficulty or ease of purchasing high-quality vegetable oil. Generally, the easier consumers perceive it to buy high-quality vegetable oil, the fewer expected barriers they have, the greater the perceived behavioral control, and the greater the likelihood of willingness to pay a premium. Environmental endowment refers to the impact of the environment on consumers’ perceived behavioral control. Therefore, this study proposes the following hypothesis:

**Hypothesis (H5).** 
*Environmental endowment has a positive influence on the willingness to pay a premium for high-quality vegetable oil.*


#### 2.2.6. Behavioral Attitude

Behavioral attitude refers to an individual’s attitude and opinion towards a specific behavior or event [39]. It consists of cognitive, affective, and behavioral aspects, reflecting an individual’s psychological state and behavioral tendencies [48]. According to the Theory of Planned Behavior, consumers’ behavioral attitude towards high-quality vegetable oil refers to the degree of liking or disliking of high-quality vegetable oil. Generally, the more positive consumers’ attitude towards high-quality vegetable oil, the stronger their willingness to pay a premium for it. Conversely, the more negative consumers’ attitude towards high-quality vegetable oil, the weaker their willingness to pay a premium for it. This study proposes the following hypothesis:

**Hypothesis (H6).** 
*Behavioral attitude has a positive influence on the willingness to pay a premium for high-quality vegetable oil.*


## 3. Materials and Methods

### 3.1. Measures

Compared to other research methods, Structural Equation Modeling (SEM) has significant advantages. It is generally believed that abstract variables cannot be directly observed, yet SEM can use a series of directly observable variables to reveal latent variables that are difficult to measure directly. Moreover, this model can simultaneously handle multiple dependent variables, thus accurately estimating the complex relationships among factors and their structures. It is worth mentioning that SEM also allows for the consideration of measurement errors in the analysis, making the research results more reliable and accurate. Therefore, in this study, SEM is employed to analyze the willingness to pay a premium for high-quality vegetable oil among urban residents and its influencing factors. The latent variables include health attitude, nutritional safety, normative beliefs, compliance motivation, consumer endowment, environmental endowment,, with each latent variable divided into three observable variables (Figure 1).

The questionnaire consists of four parts in specific. The first part of the questionnaire covers the personal and family characteristics of the respondents, including gender, age, education level, province, occupation, monthly income, health condition, and household size. The second part focuses on consumers’ cognitive level about vegetable oil.

The third and fourth parts aim to collect potential variables, including consumers’ willingness to pay a premium for high-quality vegetable oil and their purchasing behavior. In the context of the Theory of Planned Behavior framework, and building upon prior survey research, we conducted a thorough analysis of the factors influencing individuals’ willingness to pay a premium for high-quality vegetable oil. A total of 8 latent variables and 24 observable variables were identified, with specific questions and value descriptions meticulously outlined in Table 1 of the questionnaire. All variables were measured using Likert scale questions on a scale from 1, representing “strongly disagree”, to 5, representing “strongly agree”.

### 3.2. Data Collection

Before the formal survey, we randomly select 30 respondents in Beijing for a preliminary survey. Based on the preliminary survey results, adjustments were made to the questionnaire items after thorough discussions with supervisors and experts to improve the overall logic and wording of the questions.

In determining the minimum sample size for this study, based on an allowable error of 3% and a confidence level of 90%, rigorous calculations indicated that a sample size of 670 units was required (Walter et al., 1998) [52]. The formal survey was conducted in the form of a questionnaire from March to April 2023 with a combination of online and offline sampling methods. Our surveys primarily used two methods to collect questionnaire data. Firstly, we commissioned the Wen Juan Xing website (https://www.wjx.cn, accessed on 27 March 2023) and Da Ying Jia Intelligent Dietary Management System to conduct surveys targeting urban consumers in 30 provinces of China (excluding Tibet), with 35–40 samples drawn from each province and a total of 1080 questionnaires collected. After excluding invalid questionnaires, 1009 valid questionnaires were obtained. In addition, offline field research was conducted in Beijing and Hebi City, Henan Province, with a total of 100 questionnaires collected, of which 89 were valid. In total, 1098 questionnaires were collected online and offline, and about 93.05% responded.

Our study does not necessitate additional approval from an ethics committee as it does not involve animal or human clinical trials and is not unethical. Prior to completing the questionnaires, all participants provided informed consent. The information provided by participants is strictly used for scientific research purposes and will not be disclosed for any other purpose, with stringent measures in place to ensure the confidentiality of personal data. Participation in the study is entirely voluntary.

### 3.3. Description of Sample Characteristics Distribution

In terms of gender distribution, there were 481 male respondents, accounting for 43.8%, and 617 female respondents, accounting for 56.1%. The higher number of female respondents is related to women being the primary purchasers of oil in households. In traditional family roles, women often take on more responsibilities in household affairs such as cooking and food preparation. The age distribution of the respondents was relatively evenly spread across different age groups, representing urban residents of various ages in their consumption of vegetable oil. The respondents generally had a higher level of education, enabling a better understanding of the issues covered in this survey. The proportion of respondents with a bachelor’s degree or college degree was the highest at 77.9%, while those with primary school education or below had the smallest proportion at 1.5% of the total sample. There was a significant variation in personal income levels, with the highest proportion in the CNY 50,000–100,000 range at 29.4% of the total sample. This was followed by the ranges of CNY 100,001–150,000 and CNY 10,001–15,000, accounting for 22.1% and 21.9%, respectively. The samples were evenly distributed across different regions, with 30–40 urban residents randomly selected for the survey in each of the 30 provinces in China (excluding Tibet) (Table 2).

## 4. Results

### 4.1. Consumer Preferences

When consumers choose high-quality vegetable oil, they go through a selection process. Therefore, when discussing the quality attributes of high-quality vegetable oil in consumers’ minds, it is necessary to select the attributes most relevant to the quality of vegetable oil as representatives. However, it is important to avoid selecting too many attributes that make the questionnaire difficult to answer, which could affect the accuracy of the result analysis. This study reviewed the relevant literature and found that attributes such as color, transparency, taste and flavor, nutritional value, brand reputation, variety, GMO status, quality safety, and quality grade have a positive impact on consumers’ willingness to pay for vegetable oil. Due to the numerous factors affecting consumers’ willingness to pay for vegetable oil, this study only focuses on the quality attributes of vegetable oil. Therefore, this study selected 9 representative quality attributes of vegetable oil, including color, transparency, taste and flavor, nutritional value, brand reputation, variety, GMO status, quality safety, and quality grade (Table 3).

Firstly, consumers prioritize the quality safety attribute of high-quality vegetable oil the most. In total, 81.7% of consumers believe that the quality safety of high-quality vegetable oil is extremely important. This indicates that consumer safety is the primary concern for consumers, and quality safety is the bottom line for high-quality vegetable oil. Only when high-quality vegetable oil meets food safety standards can other quality attributes be discussed, which is consistent with the findings of other studies.

Secondly, consumers place a high emphasis on the nutritional value attribute, with 67.1% of samples considering the nutritional value of high-quality vegetable oil to be extremely important, following closely behind quality safety. This suggests that most consumers believe that high-quality vegetable oil should have high nutritional value. Thirdly, consumers value taste and flavor highly, with 57.8% of samples considering it to be extremely important and 89.6% considering it to be very important or important. Nearly nine out of ten consumers believe that high-quality vegetable oil should have a good taste, indicating that consumers prioritize rich and delicious taste when consuming vegetable oil. Lastly, consumers prioritize quality grade after taste and flavor, with 52.6% of samples considering quality grade to be extremely important and 40.9% considering it to be important. This means that over ninety percent of consumers believe that high-quality vegetable oil should meet high-quality grade standards, showing that consumers value national standards and use them as an essential criterion for determining high-quality vegetable oil.

In addition, consumers show relatively lower preferences for the other three items. The importance attached to the GMO status is moderate, with 43.8% of the samples considering it to be of moderate importance and 26.4% considering it to be of relatively low importance. This indicates that while most residents currently place some importance on whether vegetable oil is genetically modified, the level of concern is not high, and they are more focused on other aspects such as the nutritional qualities of the oil. Consumers’ emphasis on brand reputation is lower than that on GMO status, indicating that consumers have a moderate level of concern for brand reputation. The importance given to transparency and color is similar, suggesting that consumers do not place high importance on the appearance of the oil but rather prioritize the nutritional and other health values. This reflects a trend towards rational consumer purchasing behavior in the vegetable oil market. Consumers do not place a high emphasis on variety and type, which goes against expectations. This may be because consumers prefer a variety of oils and do not place high importance on whether the variety is of high quality.

Ranked by importance, consumers evaluate the standards for high-quality vegetable oil in the following order: quality safety, nutritional value, taste and flavor, quality grade, GMO status, brand reputation, transparency, color, and variety. It can be seen that consumers consider high-quality vegetable oil to have high safety standards, a high-quality grade, rich nutritional value, good taste and flavor, non-GMO status, a good brand reputation, good transparency and color, and a good variety.

### 4.2. The Driving Factors for Purchasing High-Quality Vegetable Oil

#### 4.2.1. Reliability Analysis

In this study, the main factors were measured using a five-point scale, so it is crucial to verify the data quality of the measurement results to ensure the meaningfulness of subsequent analysis. Firstly, the internal consistency of each dimension was analyzed using Cronbach’s alpha reliability test. The Cronbach’s alpha coefficient ranges from 0 to 1, with higher values indicating higher reliability. It is generally considered that a reliability coefficient below 0.6 is not reliable and may require redesigning the questionnaire or collecting data again for further analysis. Reliability coefficients between 0.6 and 0.7 are considered reliable, 0.7 and 0.8 are fairly reliable, 0.8 and 0.9 are very reliable, and 0.9 and 1.0 are extremely reliable [53].

The results of the reliability analysis in this analysis are shown in Table 4, with reliability coefficients for each dimension ranging from 0.7 to 1. This indicates that the scales used in this study have good internal consistency and reliability.

#### 4.2.2. Validity Analysis

According to the model fit test results in Table 5, it can be seen that CMIN/DF (chi-square degrees of freedom ratio) = 3.43 falls within the range of 3–5, and RMSEA (root mean square error of approximation) = 0.047, within the excellent range of <0.05 [54]. Additionally, the test results for TLI and CFI both reached excellent levels of 0.9 or above. Therefore, based on the comprehensive analysis results, it can be concluded that the CFA model is a good fit.

Under the premise of good model fit of the CFA model in the scale, the convergent validity (AVE) and composite reliability (CR) of each dimension of the scale will be further examined. The testing process involves calculating the standardized factor loadings of each measurement item on the corresponding dimension through the established CFA model. Then, the values of convergent validity and composite reliability for each dimension are calculated using the formulas for AVE and CR. According to the standards, the AVE value should reach a minimum of 0.5, and the CR value should reach a minimum of 0.7 to indicate good convergent validity and composite reliability [54].

Based on the analysis results in Table 6, it can be seen that in the validity testing of the scale in this study, the AVE values of each dimension all exceeded 0.5, and the CR values all exceeded 0.7. It can be concluded that each dimension has good convergent validity and composite reliability.

According to the analysis results in Table 7, it can be observed that in the discriminant validity testing in this study, the standardized correlation coefficients between each pair of dimensions are all less than the square root of the AVE value corresponding to the dimension. Therefore, it indicates that there is good discriminant validity among the various dimensions.

#### 4.2.3. Descriptive Statistics and Normality Test

Table 8 presents the results of the descriptive statistics and normality test of the factors used in this study. According to the analysis of descriptive statistics, it can be seen that the mean scores of each variable fall between 3 and 4. The scoring method of the scale is 1–5 with positive scoring, indicating that the cognitive and behavioral levels of the study participants in high-quality vegetable oil are above the medium level. The normality test of each measurement item is conducted using skewness and kurtosis, where skewness reflects the asymmetry of the residual distribution. According to the standard proposed by Kline [55], absolute values of skewness coefficients within 3 and kurtosis coefficients within 8 suggest that the data meet the requirements of approximate normal distribution. Based on the analysis results in Table 8, it can be observed that the absolute values of skewness and kurtosis coefficients of each measurement item in this study fall within the standard range, indicating that the data of each measurement item approximately satisfies the normal distribution.

### 4.3. Hypothesis Testing Results of the SEM Model Path Relationships

The theoretical model of urban residents’ willingness to pay a premium for high-quality vegetable oil proposed in this study has been confirmed. From the empirical results, it can be seen that urban residents’ willingness to pay a premium for high-quality vegetable oil is influenced by health value, monetary value, normative beliefs, environmental endowment, and behavioral attitudes, with the following impact strengths: normative beliefs (0.381) > monetary value (0.229) > behavioral attitudes (0.201) > environmental endowment (0.151) > health value (0.139) > compliance motivation (0.088). This also indicates that the indirect impact of normative beliefs and monetary value on the willingness to pay a premium should not be ignored.

#### 4.3.1. Perceived Value

Perceived value of willingness to pay a premium and its influencing factors: The perceived value of urban residents has a positive impact on their willingness to pay a premium. The empirical results of the model show that urban residents’ willingness to pay a premium for high-quality vegetable oil is jointly influenced by health value and monetary value, both significantly at the 1% level, with path coefficients of 0.139 and 0.229, respectively. This indicates that in the formation process of urban residents’ willingness to pay a premium for high-quality vegetable oil, this willingness is jointly influenced by health value and monetary value, with monetary value being the main factor in forming the perceived value of urban residents. Hypotheses H1 and H2 are supported.

The latent variable influencing the monetary value of urban residents is significant at the 1% level, with all three observed indicators of monetary value loading exceeding 0.5. Among them, the loadings of “High-quality vegetable oil are worth it” (MV2) and “If I wanted to, I could afford to spend on high-quality vegetable oil” (MV3) exceed 0.7 (as shown in the Table 9). This further indicates that monetary value plays a significant role and influences urban residents’ willingness to pay a premium for high-quality vegetable oil. “High-quality vegetable oils are worth it” (MV2) and “If I wanted to, I could afford to spend on high-quality vegetable oil” (MV3) are the main factors for urban residents in forming monetary value.

#### 4.3.2. Subjective Behavioral Norms

Subjective behavioral norms of willingness to pay a premium and its influencing factors: Urban residents’ subjective behavioral norms have a positive impact on their willingness to pay a premium. The empirical results of the model show that the willingness of urban residents to pay a premium for high-quality vegetable oil is influenced by the observed variables of normative beliefs (*p* < 0.001), with a path coefficient of 0.381. Compliance motivation does not significantly affect the willingness to pay a premium (*p* > 0.001). This indicates that in the process of forming the willingness of urban residents to pay a premium for high-quality vegetable oil, urban residents are influenced by normative beliefs, with normative beliefs being the main factor in forming the subjective behavioral norms of urban residents. Hypothesis H3 is supported, while Hypothesis H4 is not supported.

The latent variable influencing the normative beliefs of urban residents is significant at the 1% level, with all three observed indicators of normative beliefs loading exceeding 0.7. Among them, the loadings of the three observed indicators “My family supports me in spending a higher price to buy high-quality vegetable oil” (N1), “My relatives and friends think it is wise for me to spend a high price to buy high-quality vegetable oil” (N2), and “My neighbors and colleagues believe it is wise for me to spend a high price to buy high-quality vegetable oil” (N3) are 0.71, 0.769, and 0.717, respectively.

#### 4.3.3. Perceived Behavioral Control

Perceived behavioral control of willingness to pay a premium and its influencing factors: Urban residents’ perceived behavioral control has a positive impact on their willingness to pay a premium. The empirical results of the model show that urban residents’ willingness to pay a premium for high-quality vegetable oil is influenced by the observed variables of environmental endowment (*p* < 0.001), with a path coefficient of 0.151. This indicates that in the formation process of urban residents’ willingness to pay a premium for high-quality vegetable oil, this willingness is influenced by environmental endowment, with environmental endowment being the main factor in forming urban residents’ perceived behavioral control. Hypothesis H5 is supported.

The latent variable influencing the perceived behavioral control of urban residents is significant at the 1% level, with all three observed indicators of environmental endowment loading exceeding 0.5. Among them, the loadings of the observed indicators “It is very convenient for me to purchase high-quality vegetable oil” (I1) and “I know exactly where to buy high-quality vegetable oil” (I2) exceeds 0.7. This further indicates that environmental endowment plays a significant role and influences urban residents’ willingness to pay a premium for high-quality vegetable oil. “It is very convenient for me to purchase high-quality vegetable oil” (I1) and “I know exactly where to buy high-quality vegetable oil” (I2) are the main factors in the formation of urban residents’ environmental endowment.

#### 4.3.4. Behavioral Attitudes

Behavioral attitudes of willingness to pay a premium and its influencing factors: Urban residents’ behavioral attitudes have a positive impact on their willingness to pay a premium. The empirical results of the model show that urban residents’ willingness to pay a premium for high-quality vegetable oil is influenced by the observed variables of behavioral attitudes (*p* < 0.001), with a path coefficient of 0.201. This indicates that in the formation process of urban residents’ willingness to pay a premium for high-quality vegetable oil, this willingness is influenced by behavioral attitudes, supporting Hypothesis H6.

The latent variable influencing the behavioral attitudes of urban residents is significant at the 1% level, with all three observed indicators of behavioral attitudes loading exceeding 0.5. Among them, the loadings of the observed indicators “I enjoy consuming high-quality vegetable oil” (A2) is 0.74, and “I trust the quality and safety of high-quality vegetable oil” (A3) is 0.861, both exceeding 0.7. This further indicates that behavioral attitudes play a significant role and influence urban residents’ willingness to pay a premium for high-quality vegetable oil. “I enjoy consuming high-quality vegetable oil” (A2) and “I trust the quality and safety of high-quality vegetable oil” (A3) are the main factors in forming the behavioral attitudes of urban residents. Table 9 shows the convergent validity and composite reliability tests for each dimension of the scale.

## 5. Discussion

The first step of this study is to determine the main attributes of high-quality vegetable oils that consumers perceive. Consumer decisions regarding food are far from simple, as they are determined by the complex interaction of multiple factors. While food is crucial for human well-being, its importance varies from person to person, reflecting their unique values and preferences.

When evaluating the quality attributes of high-quality vegetable oils, consumers place the greatest emphasis on safety quality, nutritional value, and taste/flavor. This is in line with the research by Cao [9,10,11], de la Mata-Espinosa et al., and Huang et al. on safety quality, and Zhao [56] on nutritional value. Cao et al. [9] regard the safety of vegetable oil as the most critical quality aspect, while Zhao et al. believe that nutritional value is an essential attribute of vegetable oil. Consumers also highly value taste/flavor, which aligns with previous research indicating that most consumers seem to prefer vegetable oils with a better taste. In summary, our study findings indicate that consumers show significant interest in the safety quality, nutritional value, and taste/flavor attributes of vegetable oils [57,58,59].

The second step of this study is to determine consumers’ willingness to pay a premium for high-quality vegetable oils and its influencing factors. Consumer willingness to pay a premium for food is influenced by a combination of factors. The research findings confirmed hypotheses H1 and H2, indicating that consumers’ health values and monetary values significantly influence their positive attitudes towards high-quality vegetable oils. This is consistent with the findings of Haemoon, Shuhaiber, and Mahon [32,40,42].

Additionally, hypothesis H3 was also confirmed. In terms of normative beliefs, when urban residents perceive stronger support from those around them, especially family and relatives, for paying a premium for high-quality vegetable oils, the subjective norms built upon this basis are more likely to promote the formation of the willingness to pay a premium. This view is consistent with the results argued by Scalco [44]. Furthermore, Zagata [45] found that normative beliefs strengthen consumers’ willingness to purchase food, with the most critical source being the family, such as partners or other family members.

Hypotheses H5 and H6 were also confirmed, indicating that environmental endowments can affect the formation of willingness to pay a premium for high-quality vegetable oils. Among these, the convenience of purchasing high-quality vegetable oils and clarity on where to buy them have the most significant impact on urban residents, and the more explicit urban residents are in their preference for high-quality vegetable oils, the stronger their willingness to pay a premium. This finding is consistent with the study by Rao [39]. Consumers perceive fewer anticipated barriers and greater environmental endowments when it is easier to purchase high-quality vegetable oils in their surroundings, increasing the likelihood of their willingness to pay a premium for high-quality vegetable oils. In addition, the more positive consumers are towards high-quality vegetable oils, the stronger their willingness to pay a premium for them.

Most scholars primarily use Structural Equation Modeling for analyzing residents’ willingness to pay, behaviors, and satisfaction towards food; for example, M.L. Mitterer-Daltoé [20] utilized Structural Equation Modeling (SEM) to elucidate the relationship between attitudes, subjective norms, and past experiences on the fish consumption behavior of Brazilians. Shu-Yen Hsu [21] and colleagues applied Structural Equation Modeling to demonstrate that subjective knowledge of organic food, health consciousness, and concerns about food safety have a positive and significant impact on the willingness to purchase organic food, while fewer scholars focus on the willingness to pay a premium for edible vegetable oil.

Furthermore, the majority of studies concentrate on the willingness to pay for organic products, lacking analysis of the quality and price of edible vegetable oils. This study, based on national standards and foreign standards for edible vegetable oils, synthesizes evaluation indicators from existing studies, explores consumer perceptions of high-quality vegetable oils, constructs a model of factors influencing willingness to pay a premium for edible vegetable oil, and conducts a survey analysis of consumers’ willingness to pay a premium for and the payment levels of edible vegetable oil. This research aims to provide a reference for the pricing and consumption of edible vegetable oils.

## 6. Conclusions and Future Scope

### 6.1. Conclusions

Empirical results reveal that (1) urban residents in our country perceive high-quality vegetable oils as possessing attributes such as high safety quality, nutritional value, and grade, as well as rich in taste and flavor, a strong brand reputation, good transparency, and a variety of colors and types. (2) The health value, monetary value, normative beliefs, and attitudes of urban residents have a positive and significant impact on their willingness to pay a premium for high-quality vegetable oils, with the influences of normative beliefs and monetary value being particularly strong.

### 6.2. Recommendation

Based on the above conclusions, consumer willingness to pay a premium for high-quality vegetable oil can be enhanced through three intervention paths:(1)Quality and safety are the most important quality attributes that consumers are concerned about when it comes to vegetable oil. Enterprises must strictly adhere to hygiene, quality, and safety standards in the production process to ensure the safety and reliability of the products.(2)Strengthening publicity and popularization to enhance familiarity and preference for high-quality vegetable oil, cultivating positive behavioral attitudes among consumers.(3)Providing marketing promotional opportunities and enhancing regional marketing of high-quality vegetable oil to optimize consumers’ environmental endowment and strengthen perceived behavioral control.

### 6.3. Limitations and Future Scope

This study is not without limitations. Due to the limitations of cost and time, this study focuses only on the consumption of edible vegetable oils among urban residents and does not investigate rural areas. Future research could include the edible vegetable oil consumption of rural consumers to more comprehensively assess the situation of edible vegetable oil consumption in China. Additionally, this research excludes individuals below 18 years of age and those with an education level above the national average. Therefore, future studies could pay more attention to the population under 18 and those with lower educational levels to uncover the preferences for edible vegetable oils among these specific groups.

## Figures and Tables

**Figure 1 foods-13-01168-f001:**
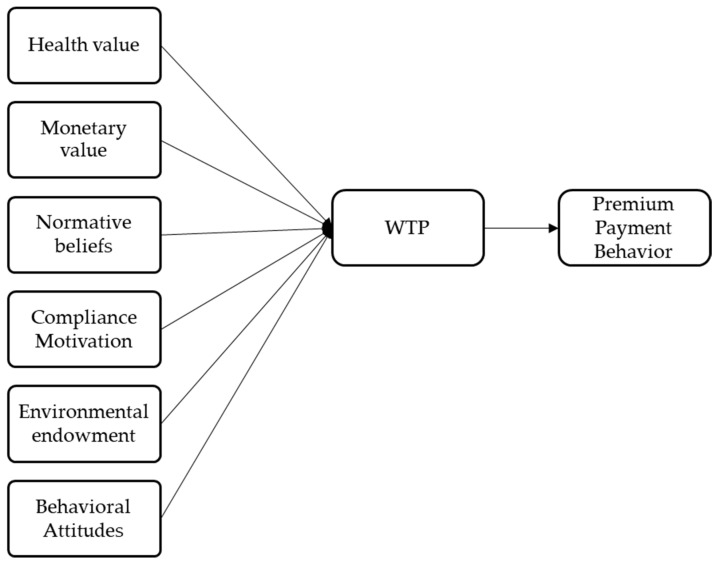
Structural Equation Model.

**Table 1 foods-13-01168-t001:** Variable selection.

Variables.	Items	References
Health Value	PV1: Eating high-quality vegetable oil is beneficial to health.	[20]
	PV2: High-quality vegetable oil has higher nutritional value.	
	PV3: High-quality vegetable oil is safer.	
Monetary Value	MV1: High-quality vegetable oil is not expensive.	[49]
	MV2: High-quality vegetable oils are worth it.	
	MV3: If I wanted to, I could afford to spend on high-quality vegetable oil.	
Behavioral Attitudes	A1: I will purchase high-quality vegetable oil based on the packaging.	[48]
	A2: I enjoy consuming high-quality vegetable oil.	
	A3: I trust the quality and safety of high-quality vegetable oil.	
Normative Beliefs	N1: My family supports me spending a higher price to buy high-quality vegetable oil.	[50]
	N2: My relatives and friends believe it is wise for me to spend a higher price to buy high-quality vegetable oil.	
	N3: My neighbors and colleagues think it is wise for me to spend a higher price to buy high-quality vegetable oil.	
Compliance Motivation	C1: I will take my family’s opinion into consideration.	[46]
	C2: I will take my relatives’ and friends’ opinions into consideration.	
	C3: I will take my neighbors’ and colleagues’ opinions into consideration.	
Environmental Endowment	I1: It is very convenient for me to purchase high-quality vegetable oil.	[29,51]
	I2: It is very convenient for me to purchase high-quality vegetable oil.	
	I3: High-quality vegetable oil is not expensive, so you can buy it if you want.	
WTP	B1: I am willing to pay a higher price for higher-quality edible vegetable oil.	
	B2: I plan to choose high-quality vegetable oil more in the future.	
	B3: I am willing to recommend others to buy high-quality vegetable oil.	
Premium Payment Behavior	P1: I often spend a higher price to buy high-quality vegetable oil.	
	P2: I often consume high-quality vegetable oil.	
	P3: I often recommend others to eat high-quality vegetable oil.	

**Table 2 foods-13-01168-t002:** Description of sample characteristics distribution.

Variables	Items	Samples	Percentage (%)
Gender	Male	481	43.8
	Female	617	56.1
Age	0–25 years old	209	19.0
	26–35 years old	290	26.4
	36–45 years old	248	22.5
	46–55 years old	224	20.4
	More than 56 years old	127	11.5
Marriage	Married	845	77.0
	Single	242	22.0
	Divorced	8	0.7
	Widowed	3	0.3
Occupation	Full-time employment	941	85.7
	Internship	23	2.1
	Retired	40	3.6
	Student	77	7.0
	Unemployed	17	1.5
Education	Primary school or below	16	1.5
	Junior school	47	4.3
	Senior school	116	10.6
	College/university	855	77.9
	Master’s or above	64	5.8
Yearly Personal Income	Less than CNY 10,000 (1428 dollars)	94	8.6
	CNY 10,000–50,000 (1428–7143 dollars)	240	21.9
	CNY 50,001–100,000 (7143–14,280 dollars)	323	29.4
	CNY 100,001–150,000 (14,280–21,430 dollars)	243	22.1
	CNY 150,001 (21,430 dollars) and above	198	18.0

**Table 3 foods-13-01168-t003:** Descriptive statistical analysis of preferences for high-quality vegetable oil attributes.

Variables	Variable Assignment	Samples	Percentage (%)
Color	Not important = 1	2	0.2
	Slightly unimportant = 2	46	4.2
	Moderate = 3	225	20.5
	Comparatively important = 4	661	60.2
	Extremely important = 5	164	14.9
Transparency	Not important = 1	2	0.2
	Slightly unimportant = 2	38	3.5
	Moderate = 3	300	27.3
	Comparatively important = 4	483	44.0
	Extremely important = 5	275	25.0
Taste and Flavor	Not important = 1	2	0.2
	Slightly unimportant = 2	15	1.4
	Moderate = 3	97	8.8
	Comparatively important = 4	349	31.8
	Extremely important = 5	635	57.8
Nutritional Value	Not important = 1	0	0.0
	Slightly unimportant = 2	9	0.8
	Moderate = 3	67	6.1
	Comparatively important = 4	285	26.0
	Extremely important = 5	737	67.1
Brand Reputation	Not important = 1	4	0.4
	Slightly unimportant = 2	46	4.2
	Moderate = 3	230	20.9
	Comparatively important = 4	521	47.4
	Extremely important = 5	297	27.0
Variety	Not important = 1	8	0.7
	Slightly unimportant = 2	64	5.8
	Moderate = 3	290	26.4
	Comparatively important = 4	481	43.8
	Extremely important = 5	255	23.2
GMO Status	Not important = 1	12	0.7
	Slightly unimportant = 2	58	5.8
	Moderate = 3	192	26.4
	Comparatively important = 4	336	43.8
	Extremely important = 5	500	23.2
Quality Safety	Not important = 1	1	0.1
	Slightly unimportant = 2	8	0.7
	Moderate = 3	31	2.8
	Comparatively important = 4	161	14.7
	Extremely important = 5	897	81.7
Quality Grade	Not important = 1	2	0.2
	Slightly unimportant = 2	10	0.9
	Moderate = 3	60	5.5
	Comparatively important = 4	449	40.9
	Extremely important = 5	577	52.6

**Table 4 foods-13-01168-t004:** Reliability analysis of the scale.

Latent Variables	Cronbach’s Alpha	Measurable Variables
Health Value	0.805	3
Monetary Value	0.773	3
Behavioral Attitudes	0.772	3
Normative Beliefs	0.775	3
Compliance Motivation	0.823	3
Environmental Endowment	0.736	3
Willingness to Pay (WTP)	0.705	3
Premium Payment Behavior	0.735	3

**Table 5 foods-13-01168-t005:** Model fit test.

Indicator	Standards	Result	Assessment
CMIN/DF	<5	3.43	Passed
RMSEA	<0.10	0.047	Passed
IFI	>0.9	0.946	Passed
TLI	>0.9	0.929	Passed
CFI	>0.9	0.946	Passed

The data are adapted from the author (2023). Note that CMIN/DF = chi-square/degrees of freedom; TLI = Tucker–Lewis Index; CFI = Comparative Fit Index; IFI = Incremental Fit Index; RMSEA = Root Mean Square Error of Approximation.

**Table 6 foods-13-01168-t006:** Convergent validity and composite reliability test of various dimensions of the scale/aggregate validity.

Path Relationship			Estimate	AVE	CR
PV1	<---	Health Value	0.817	0.5831	0.807
PV2	<---	Health Value	0.709		
PV3	<---	Health Value	0.761		
A2	<---	Behavioral Attitudes	0.853	0.5509	0.7832
A3	<---	Behavioral Attitudes	0.746		
A4	<---	Behavioral Attitudes	0.607		
MV1	<---	Monetary Value	0.593	0.5555	0.7861
MV2	<---	Monetary Value	0.833		
MV3	<---	Monetary Value	0.788		
N1	<---	Normative Beliefs	0.707	0.5631	0.7759
N2	<---	Normative Beliefs	0.769		
N3	<---	Normative Beliefs	0.719		
C1	<---	Compliance Motivation	0.678	0.6197	0.8285
C2	<---	Compliance Motivation	0.843		
C3	<---	Compliance Motivation	0.83		
I1	<---	Environmental Endowment	0.846	0.5236	0.7615
I2	<---	Environmental Endowment	0.752		
I3	<---	Environmental Endowment	0.538		

**Table 7 foods-13-01168-t007:** Correlation coefficients alongside the square root of AVE.

Dimensions	Health Value	Monetary Value	Behavioral Attitudes	Normative Beliefs	Compliance Motivation	Environmental Endowment
Health value	0.583					
Monetary value	0.372	0.551				
Behavioral attitudes	0.501	0.356	0.556			
Normative beliefs	0.392	0.464	0.548	0.563		
Compliance motivation	0.228	0.289	0.24	0.472	0.620	
Environmental endowment	0.414	0.438	0.333	0.407	0.233	0.524
The square root of the AVE	0.769	0.742	0.745	0.750	0.787	0.724

Note: the *p*-value is less than 0.05.

**Table 8 foods-13-01168-t008:** Descriptive statistics and normality test results of each dimension and measurement item.

Dimensions	Item	Mean	SD	Skewness	Kurtosis	Population (M)	Population (SD)
Health Value	PV1	4.32	0.687	−0.761	0.392	4.3145	0.6024
	PV2	4.28	0.720	−0.688	−0.068		
	PV3	4.34	0.723	−0.824	0.075		
Behavioral Attitudes	A2	4.20	0.793	−0.719	−0.047	4.1044	0.7019
	A3	4.20	0.826	−0.753	−0.146		
	A4	3.91	0.917	−0.391	−0.668		
Monetary Value	MV1	3.33	0.958	0.003	−0.763	3.6269	0.7515
	MV2	3.70	0.869	−0.511	0.002		
	MV3	3.86	0.888	−0.555	−0.041		
Normative Beliefs	N1	3.92	0.897	−0.495	−0.277	3.7365	0.7523
	N2	3.66	0.908	−0.215	−0.505		
	N3	3.63	0.913	−0.246	−0.381		
Compliance Motivation	C1	3.90	0.873	−0.643	0.261	3.7046	0.7608
	C2	3.68	0.886	−0.361	−0.194		
	C3	3.54	0.891	−0.248	−0.285		
Environmental endowment	I1	4.04	0.798	−0.734	0.584	3.9575	0.6864
	I2	4.14	0.815	−0.840	0.523		
	I3	3.69	0.926	−0.367	−0.408		
WTP	B1	3.86	0.893	−0.541	−0.124	3.9341	0.6778
	B2	4.03	0.803	−0.621	0.316		
	B3	3.91	0.867	−0.638	0.327		
Premium Payment Behavior	P1	3.78	0.951	−0.532	−0.177	3.7714	0.776
	P2	3.89	0.919	−0.650	0.129		
	P3	3.65	1.006	−0.467	−0.239		

**Table 9 foods-13-01168-t009:** Convergent validity and composite reliability test for each dimension.

Path Relationship			Estimate	S.E.	C.R.	*p*
WTP	<---	Health Value	0.139	0.036	4.019	***
WTP	<---	Monetary Value	0.229	0.028	6.635	***
WTP	<---	Normative Beliefs	0.381	0.04	8.34	***
WTP	<---	Compliance Motivation	0.088	0.024	2.883	0.004
WTP	<---	Environmental Endowment	0.151	0.039	4.454	***
WTP	<---	Behavioral Attitudes	0.201	0.04	5.268	***
Premium payment behavior	<---	WTP	0.874	0.059	17.131	***
PV1: Eating high-quality vegetable oil is beneficial to health.	<---	Health Value	0.811	0.044	23.189	***
PV2: High-quality vegetable oil has higher nutritional value.	<---	Health Value	0.714	0.043	21.52	***
PV3: High-quality vegetable oil is safer.	<---	Health Value	0.763			
MV1: High-quality vegetable oil is not expensive.	<---	Monetary value	0.591	0.044	18.291	***
MV2: High-quality vegetable oils are worth it.	<---	Monetary value	0.825	0.044	23.298	***
MV3: If I wanted to, I could afford to spend on high-quality vegetable oil.	<---	Monetary value	0.796			
N1: My family supports me spending a higher price to buy high-quality vegetable oil.	<---	Normative Beliefs	0.71	0.048	20.216	***
N2: My relatives and friends believe it is wise for me to spend a higher price to buy high-quality vegetable oil.	<---	Normative Beliefs	0.769	0.05	21.387	***
N3: My neighbors and colleagues think it is wise for me to spend a higher price to buy high-quality vegetable oil.	<---	Normative Beliefs	0.717			
C1: I will take my family’s opinion into consideration.	<---	Compliance Motivation	0.678	0.036	22.326	***
C2: I will take my relatives’ and friends’ opinions into consideration.	<---	Compliance Motivation	0.843	0.039	26.014	***
C3: I will take my neighbors’ and colleagues’ opinions into consideration.	<---	Compliance Motivation	0.83			
I1: It is very convenient for me to purchase high-quality vegetable oil.	<---	Environmental Endowment	0.844	0.083	16.212	***
I2: I know exactly where to buy high-quality vegetable oil.	<---	Environmental Endowment	0.752	0.076	16.117	***
I3: High-quality vegetable oil is not expensive, so you can buy it if you want.	<---	Environmental Endowment	0.54			
B1: I am willing to pay a higher price for higher-quality edible vegetable oil.	<---	WTP	0.643			
B2: I plan to choose high-quality vegetable oil more in the future.	<---	WTP	0.719	0.052	19.396	***
B3: I am willing to recommend others to buy high-quality vegetable oil.	<---	WTP	0.598	0.054	16.757	***
P1: I often spend a higher price to buy high-quality vegetable oil.	<---	Premium Payment Behavior	0.694			
P2: I often consume high-quality vegetable oil.	<---	Premium Payment Behavior	0.737	0.051	20.167	***
P3: I often recommend others to eat high-quality vegetable oil.	<---	Premium Payment Behavior	0.656	0.054	18.433	***
A1: I will purchase high-quality vegetable oil based on the packaging.	<---	Behavioral Attitudes	0.861	0.065	18.977	***
A2: I enjoy consuming high-quality vegetable oil.	<---	Behavioral Attitudes	0.74	0.061	18.251	***
A3: I trust the quality and safety of high-quality vegetable oil.	<---	Behavioral Attitudes	0.603			

Note: *** indicates that the significance level is less than 0.001.

## Data Availability

The data presented in this study are available on request from the corresponding author. The data are not publicly available due to privacy restrictions.

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
