# Peer review of "Consumer Preferences and Willingness to Pay for High-Quality Vegetable Oils: A Cross-Sectional Analysis of Chinese Residents"

_foods, 2024, doi:10.3390/foods13081168_

Round 1

Reviewer 1 Report

Comments and Suggestions for Authors

The manuscript highlights important approaches and tools to boost public policies and entrepreneurs. However, in its current form, it contains some flaws that prevent a reasonable understanding of the entire work and prevent a reliable conclusion.

The biggest flaw is the poor description of the materials and methods It does not allow the understanding of the work carried out. Also, a clear separation between the material and methods section, results and discussion is lacking.

In additon, there are some statements based only on possible assumptions and a serious error between what is intended to be achieved, summarized in the title, and what the methodology actually allowed to be obtained.

 Important points that need to be reviewed are described below.

Page: 1     -  Line: 35-36

Put together both paragraphs

Page: 1      -  Line: 36

Exchange “grams” by “g” and henceforth

Page:  1     -  Line: 36-40

Insert references about the presented data

Page:  1     -  Line: 36-45

Some connection is missing at the same, at the time some degree of repetetion in those paragraphs. The information provided should be rewritten for a better understanding

Page: 2      -  Line:46-48

This sentence “The main types of edible vegetable oils consumed in China are soybean oil, rapeseed 46 oil, and peanut oil, with specialty vegetable oils accounting for a relatively low proportion. 47 Woody oils such as camellia oil and olive oil only make up 2% of the total consumption 48 of edible vegetable oils in the country” should be put together with the last paragraph

Page:  2     -  Line: 47-57

Both ideas should be presented toghether as the implications of such a overconsumption

Page:  2     -  Line: 53

Add the reference to “The "Report on Nutrition and 53 Chronic Disease Status of Chinese Residents (2020)"

Page:  2     -  Line:61-65

It was informed that the main vegetable oils consumed in China are soybean, rapessed and peanut oil. But it was stated “The World Health Organization (WHO) has released dietary guidelines based on scientific research evidence, which suggest that reducing the intake of saturated fatty acids and trans fatty acids is associated with a lower risk of cardiovascular diseases and overall mortality. It is recommended to use polyunsaturated fatty acids and monounsaturated fatty acids from plant sources.”, it does not seem related to the chinese issues, because those oil sources are mainly unsatureted and does not contain trans fat acids. This matter should be better stressed, and related not only to the cooking oil, but the overall fat consumption. In addition, other factors, such as life style, phisical activities, overall diet also play a major role in the people’s health .

Page:  2     -  Line:93-95

The sentence “In previous literature on food consumption research, the factors related to consumer 93 preferences and purchase intentions for food were mainly explored using structural equa- 94 tion modeling, including aspects such as behavior attitude, subjective behavior norms, 95 and environmental endowments.” should be placed in the beginning of the anterior paragraph (line77)

Page:  3     -  Line: 114

Use a standard word, preferable “vegetable oil” instead “plant oil”

Page: 3-4   -  Line: 141-160

The second paragraph of this section seems to suit more as the introductory idea, and then followed by the first paragraph

Page:   4    -  Line: 191

Is the sentence a new section (2.1.4) ? “

Page:  3     -  Line: 127

The section Materials and Methods is quite confusing. It should be better explained the methodological strategy applied to evaluate the hypothesis, encompassing the surveys and the statistical design. Most of those information  are in the Results section

Page:    5   -  Line: 220

The plataforma “Wen Juan Xing “ was quoted, nevertheless no link is presented to access it

Page:  5  -  Line: 229-232

There is a huge bias in this research because the sampled population does not represent the whole society groups. Therefore, the results does not fulfill the title of the manuscript (Consumer Preferences and Willingness to Pay for high-quality 2 Vegetable Oil: A Cross-sectional Analysis of Chinese Residents). Moreover, according to the statmentes of the own authors, the degree of literacy and purchase capability plays importante roles in their selection for food. Therefore, the whole work has a bias that comprimisse the discussion and conclusions

Page:  6  -  Line: table 1

Correct the word “student” in the table 1

Page:  6  -  Line: table 1

Sugestion: Exchange the currency and use UD dollars to express the incomes (it is easy for a global understanding)

Page:  7  -  Line: 260-264

According to the statements, consumers related high quality wiht unsaturated fatty acids and trace elements. Nevertheless, this information was not provided to the respondentes. It means, that authors are assuming that consumers already link quality with fatty acids profile and minor compounds. Woul it be true? Actually, the definition of quality was not given as a standard to the respondents. Please, improve the statements and discussion without assumptions.

Page:  8  -  Line: Section 3.3

It is very hard to comprehend the results and the discussions, since no explanation was given about the applied methodology to evaluate the quoted dimensions. The Pearson correlations figures are very low to state there was some actual correlation among the variables. The material and methods must be rebuilt in order to clarify how the information was acquired, before just presenting them in the results/discussion. There is no clear separation between, M&M, Results and discussion

Comments on the Quality of English Language

It meets a good standard

Reviewer 2 Report

Comments and Suggestions for Authors

Dear Authors, the manuscript is interesting; however, I suggest several improvements.

Abstract. The abstract is well-structured and provides all the necessary information.

Introduction. The issue is well presented and grounded by the statistical evidence. The previous research in the field is presented and an existing gap in literature is revealed. However, the purpose of the study is not clearly formulated. Also, the introduction is too long. I would suggest removing the presentation and analysis of the TPB and SEM (lines 66-120) from the introduction and call it “Theoretical Substantiation”.

Materials and Methods. Not all the hypotheses are grounded by the literature. For example, H2: Monetary value has a positive influence on the willingness to pay a premium for high-quality vegetable oil, needs a more proper theoretical substantiation. Subjective norms and Perceived behavioral control are not grounded by the literature at all. Therefore, the hypotheses H3-H6, even being quite clear, evoke doubts. Also, it is common providing a structural model representing the hypotheses.

Considering the Materials and Methods section, it is not finished. It lacks the information about the research methodology, questionnaire, sample, and the procedure.  

Results. The section 3.1 should be moved to the Materials and Methods part, as it does not provide research results. Also, sample characteristics like margin of error and confidence level should be indicated. The paragraph in lines 220-222 is redundant (except the effective sample rate).  

Despite that the authors present the structure of the questionnaire used, the questions and scales should be also presented for the replicability by others (in the Materials and Methods section).

The explanation of Cronbach’s alpha coefficients, AVE, CR marginal values should be grounded by literature (different explanations can be found in scientific literature).  

Discussion. The information provided in the Discussion section is NOT a discussion. It is the core result of the research. Therefore, I suggest replacing all this information to the Section 3. When it comes to a discussion, it is absent. In the discussion part, the authors are supposed to compare their results with the results obtained by the other scholars in order to demonstrate the novelty and contribution to the field. Currently this part is missing, and the contribution of this paper is not clearly revealed.

Conclusions. I suggest stressing out the limitations of the research more evidently.
